# Carotid Atherosclerosis Progression in Postmenopausal Women Receiving a Mixed Phytoestrogen Regimen: Plausible Parallels with Kronos Early Estrogen Replacement Study

**DOI:** 10.3390/biology9030048

**Published:** 2020-03-06

**Authors:** Tatiana V. Kirichenko, Veronika A. Myasoedova, Alessio L. Ravani, Igor A. Sobenin, Varvara A. Orekhova, Elena B. Romanenko, Paolo Poggio, Wei-Kai Wu, Alexander N. Orekhov

**Affiliations:** 1Research Institute of Human Morphology, 3 Tsyurupy Str., 117418 Moscow, Russia; igor.sobenin@gmail.com (I.A.S.); a.h.opexob@gmail.com (A.N.O.); 2National Medical Research Center of Cardiology, 15A 3 Cherepkovskaya Str., 121552 Moscow, Russia; 3Institute of General Pathology and Pathophysiology, 8 Baltiyskaya Str., 125315 Moscow, Russia; myika@yandex.ru (V.A.M.); varvaraao@gmail.com (V.A.O.); 4Centro Cardiologico Monzino IRCCS, Via Carlo Parea 4, 20138 Milan, Italy; alessio.ravani@ccfm.it (A.L.R.); paolo.poggio@ccfm.it (P.P.); 5Institute for Atherosclerosis Research, Skolkovo Innovative Center, 143025 Moscow, Russia; 6Department of Molecular Basis of Ontogenesis, Belozersky Institute of Physical and Chemical Biology, Moscow State University, 119234 Moscow, Russia; romanenkoeb@mail.ru; 7Department of Internal Medicine, National Taiwan University Hospital, Bei-Hu Branch, Taipei 108, Taiwan; weikaiwu0115@gmail.com

**Keywords:** women, postmenopause, cIMT progression, atherosclerosis, phytoestrogens

## Abstract

This randomized double-blinded, placebo-controlled clinical trial evaluated the progression of intima-media thickness of common carotid artery (cIMT) and the effect of phytoestrogen therapy on atherosclerosis development in early and late postmenopausal women. The 2-year cIMT progression was evaluated in 315 early postmenopausal women aged 40–55 years and in 231 late postmenopausal women aged 60–69 years free of cardiovascular disease. B-mode ultrasound was done at baseline and after 12 and 24 months of follow-up. The study revealed no significant changes in the rate of cIMT progression in 315 early postmenopausal women. By contrast, a statistically significant difference in the rate of atherosclerosis development was observed in late postmenopausal women treated with phytoestrogens compared to placebo (*p =* 0.008). The rate of cIMT progression in the placebo group was 0.019 mm/year led to a significant increase of cIMT during the observation period (*p =* 0.012), while the rate of cIMT progression in phytoestrogen late postmenopausal recipients was 0.011 mm/year, and total change did not reach statistical significance during the follow-up period (*p =* 0.101). These results suggest that late postmenopausal women can be a suitable cohort for trials assessing the anti-atherosclerosis effects of phytoestrogen preparations. In particular, the beneficial effect of phytoestrogens on cIMT progression was demonstrated in late postmenopausal women.

## 1. Introduction

The problem of atherosclerosis progression in women after menopause has been extensively studied in a number of large-scale clinical studies over the past decades [1,2,3,4,5,6]. In the industrialized countries, atherosclerosis plays a key role in the development of cardiovascular diseases that account for the largest part in the structure of mortality [7]. Atherosclerosis progression is especially prominent in late postmenopausal women. Hormone replacement therapy (HT) is widely used to restore women’s health in the late postmenopausal period, but epidemiological studies delivered controversial data on its effects on cardiovascular diseases. The possibility to reduce cardiovascular risk by means of HT is being currently explored in clinical studies [8,9]. The results of the Women’s Health Initiative (WHI) study have demonstrated a negative risk/benefit ratio of HT for cardiovascular protection in late menopausal women [10]. ELITE (Early versus Late Intervention Trial of Estradiol) study showed a significant effect of HT on carotid intima media thickness (cIMT) progression in early postmenopausal period (within 6 years after menopause) but not in late postmenopausal period (more than 10 years after menopause) [11]. One of the possible explanations would be the “timing hypothesis”, according to which HT may have multiple beneficial effects when administered during the early period after menopause, but not during the late postmenopausal period [12]. The Kronos Early Replacement Study (KEEPS) trial was designed to monitor the development of atherosclerosis in early postmenopausal women and evaluate the effectiveness of HT for carotid atherosclerosis progression in early postmenopause [13]. However, in this study, the increase of cIMT during the observation period was not significant enough to demonstrate the possible effects of HT.

Natural preparations based on botanicals are currently widely used as supporting therapy, since they usually have fewer side effects and contraindications for long-term use [14,15,16]. Such preparations have been developed for reduction of cardiovascular risk. In particular, phytoestrogens demonstrate cardioprotective effects in late postmenopausal women [16,17,18]. Several clinical trials revealed lipid-lowering [19] and hypotensive effects [20] of phytoestrogens as well as decrease of fasting glucose level in phytoestrogen recipients [21]. The aim of the present study was to evaluate cIMT progression and the effect of phytoestrogen therapy on atherosclerosis development in different periods in early and late postmenopausal women.

## 2. Materials and Methods

### 2.1. Study Design

The study was performed in accordance with the Declaration of Helsinki of 1975 and its revised version of 2013. The study protocol was approved by the Ethics Committee of the Institute for Atherosclerosis Research, Moscow (protocol No. 124-13 of 17 October 2013). All study participants provided written informed consent upon enrollment. The inclusion and observation of patients was carried out at the clinical base of the Institute for Atherosclerosis Research. The trial measuring the cIMT progression in women after menopause was specified in the protocol of the Study of Anti-atherosclerotic and Estrogen-like Activity of Karinat (ClinicalTrials.gov NCT01741974). The 2-year randomized double-blinded placebo-controlled study was conducted to estimate the effectiveness of phytoestrogen therapy on atherosclerosis progression in two subgroups of healthy women: early and late postmenopausal [22]. Natural preparation Karinat (INAT-Pharma, Russia) used in this study as a source of phytoestrogens contains in each 500 mg capsule 40 mg of grape (*Vitis vinifera*) seeds, 115 mg of green tea (*Camellia sinensis*) leaves, 160 mg of hop (*Humulus lupulus*) cones powder, 100 mg of garlic (*Allium sativum*) powder, and 85 mg of excipients. A daily dose of Karinat^®^ provides, for estimated administration, procyanidin—27.3 mg, genistein—2.5 mg, daidzein—11.8 mg, flavones—4.6 mg, resveratrol—3.5 mg, other polyphenolic compounds—44.6 mg [15]. Capsules of similar size and shape were used as placebo. Study participants were randomly divided into groups administrated three capsules of phytoestrogen preparation or placebo per day for 2 years. Measurements of cIMT were made at baseline and after 12 and 24 months of medication intake.

### 2.2. Patients

Two different subgroups of healthy women were enrolled: those in early postmenopausal and those in late postmenopausal period. Study participants from the early postmenopausal subgroup were aged between 40 and 55 years and were free of cardiovascular disease. The main inclusion criteria for early postmenopausal women were the following:age 40–55 years old at baseline;menstruation absent for at least 6 months and for no longer than 24 months;last spontaneous menstruation occurring after the age 40;plasma follicle-stimulating hormone (FSH) level >35 Ui/L and estradiol level <25 pg/mL;mammography without nodal form of mastopathy or breast cancer signs;absence of hypolipidemic therapy or HT within the last 6 months;absence of chronic diseases requiring long-term medication.

Study participants of the late postmenopausal subgroup matched the following inclusion criteria:age >60 years old at baseline;>5 years after menopause;mammography without nodal form of mastopathy or breast cancer signs;absence of hypolipidemic therapy or HT within last 6 months;absence of chronic diseases requiring long-term medication.

The exclusion criteria for both subgroups were

history or diagnostic of nodal form of mastopathy or breast cancer;history of coronary heart disease or stroke;untreated high blood pressure;obesity (BMI >30 kg/m^2^);smoking;deep vein thrombosis or pulmonary embolism;history of chronic kidney disease;history of liver dysfunction;type 2 diabetes.

Criteria for early postmenopausal group met the criteria of KEEPS protocol with minimal difference (age 42–58 years and at least 6 months but no more than 36 months after final menstrual period for KEEPS inclusion criteria and BMI >30 mm^2^/kg against 35 mm^2^/kg for KEEPS as exclusion criterion) [23]. Surgically menopaused women were not included in the study.

### 2.3. cIMT Measurement

Carotid arteries were examined by B-mode high-resolution ultrasonography using a linear array vascular probe 7.5 MHz on ultrasonic scanner SonoScape SSI-1000 (SonoScape, China). The examination included scanning of the left and right common carotid arteries, the carotid bifurcation area, as well as external and internal carotid arteries with a focus on the far wall of the artery in three fixed projections—anterior, lateral, and posterior [24,25]. Measurement of cIMT was performed at the far wall of the common carotid artery 10 mm opposite the top of the carotid bifurcation. cIMT was measured as the distance from the leading edge of the first echogenic area to the leading edge of the second echogenic area. One researcher was responsible for all cIMT measurements throughout the study. The whole procedure was recorded on a digital scan medium for subsequent analysis by independent certified reader in blinded manner using dedicated software package M’Ath 3.1 (Metris, SRL France), which allows single measurements to be taken with a precision of <0.07 mm; since the mean value of cIMT is calculated from 100 to 150 measurements on a 10 mm length of the far wall, the resulting precision of individual result accounts for <0.01 mm [26]. Reproducibility of IMT measurements was assessed according to the protocol of IMPROVE Study [27]. Within-operator coefficient of variation was 2.3% for mean сIMT; reproducibility coefficient accounted for 0.042. The average of three mean measurements (in the anterior, posterior, and lateral projections) was considered as an integral indicator of cIMT.

### 2.4. Statistical Analysis

Statistical analysis was performed using SPSS 21.0 (IBM SPSS Statistics, IBM Corp., Armonk, NY, USA). The intended target sample of early postmenopausal women was set to 300 participants (randomized 150:150), which, based on the initial statistical power estimate, would provide a 94.3% chance to show a 5% difference for cIMT change over 2 years of follow-up with α error level of 0.05. The target sample for late postmenopausal women was set to 220 participants (randomized 110:110), which would give a 97.6% chance to show a 10% difference for cIMT change over 2 years of follow-up with α error level of 0.05, but not a 5% difference in cIMT change (50.4% chance). We used the recommendations on reporting statistics by Lang and Secic [28]. For continuous variables, Kolmogorov–Smirnov test with Lilliefors’s correction was performed to estimate the data distribution. Depending of the results, parametric or non-parametric analysis was further performed. The results were expressed in terms of means and standard deviations. Significance was defined at the 95% level of confidence. The within-group progression of cIMT was assessed by linear regression analysis with age, body mass index, FSH, SBP, DBP, triglycerides, LDL and HDL cholesterol, and time after menopause used as covariates. Primary endpoints were annual rate of cIMT change and 2-year absolute cIMT change. To calculate the annual change of cIMT, the 2-year change (2-year IMT minus baseline IMT) was divided by exact number of days between visits and multiplied by 365. The ultrasound data on the differences of the mean cIMT registered at baseline and at follow-up visits between groups were analyzed by paired two-tailed *t*-test.

## 3. Results

A total of 315 early postmenopausal and 231 late postmenopausal women were recruited in the study (Appendix A). All included women had normal blood pressure and lipid profile. There were no statistically significant differences between phytoestrogen and placebo recipients of both subgroups in terms of clinical and laboratory parameters such as age, FSH level, blood pressure, BMI, levels of total cholesterol, triglycerides, HDL and LDL cholesterol. The number of current smokers was about 2% in the early postmenopausal group and less than 1% in the late postmenopausal group. Alcohol consumption was less than 1% in each group. Clinical and laboratory characteristics of early postmenopausal women at baseline are presented in Table 1, and the main characteristics of late postmenopausal women are presented in Table 2.

During the 24-month follow-up period, there were no significant differences in terms of clinical and laboratory parameters between the participants from early and late postmenopausal subgroups, treated with phytoestrogens or placebo (data not shown). A total of nine participants were excluded from the study due to non-attendance of the intermediate visit after 1 year of follow-up. No adverse effects were observed in both groups during the administration period. The overall compliance with the study drug was 93% in all groups over the whole observation period.

All study participants underwent ultrasound investigation of carotid arteries at baseline and after 12 and 24 months of follow-up. The average cIMT at baseline was 0.784 ± 0.098 mm in the overall early postmenopausal group. In the late postmenopausal group, the average cIMT at baseline was 0.844 ± 0.131 mm. The mean cIMT did not differ significantly in phytoestrogen and placebo recipients of each subgroup. The dynamics of cIMT changes after 12 and 24 months of follow-up in early and late postmenopausal subgroups is presented in Table 3 and Table 4, respectively.

No statistically significant difference of mean cIMT was observed in the early postmenopausal subgroup of participants after 12 or 24 months of follow-up, both overall and in the subgroups of phytoestrogen and placebo recipients. The annual change of mean cIMT was 0.007 mm per year in the overall group, 0.007 mm per year in the phytoestrogen group, and 0.006 mm per year in the placebo group.

In the late postmenopausal subgroup, a statistically significant cIMT increase was observed in the placebo group (*p =* 0.001). The annual change of cIMT was 0.019 mm per year in the placebo group and 0.011 mm per year in the phytoestrogen group. The difference between groups was significant (*p =* 0.021). The absolute 2-year cIMT increase was 0.037 mm in the placebo group and 0.021 mm in the phytoestrogen group, and the between-group difference was also significant (*p =* 0.014).

## 4. Discussion

The present study monitored the atherosclerosis progression in early postmenopausal women receiving the treatment with phytoestrogen preparation or placebo and found no significant increase of cIMT during the 2-year follow up. The annual rate if cIMT increase was 0.007 mm, which corresponded to the results of previous studies conducted on early postmenopausal women of matching age [29,30,31]. In particular, in the KEEPS study, the annual rate of cIMT progression was also 0.007 mm [31]. In late postmenopausal women a significant cIMT progression was observed in the subgroup receiving placebo, but not phytoestrogens, the annual rate of cIMT progression was 0.019 and 0.011 mm, respectively. Similar annual rate of cIMT progression has been shown in previous studies conducted on late postmenopausal women [32,33]. Therefore, the significant cIMT was observed only in late postmenopausal women. The different rates of cIMT progression in early and late postmenopausal groups can be explained by mechanisms of vascular aging which include chronic low-grade inflammation, oxidative stress, mitochondrial dysfunction, cellular senescence, impaired resistance to molecular stressors, genomic instability, led to age-related endothelial injury and development of atherosclerotic lesions in the arterial wall [34].

This study has its limitations. One of the drawbacks was the small sample size. Distribution of baseline сIMT values and annual progression values of сIMT in all groups was normal, but due to the fact that the sample size of the present study was small we cannot exclude that selection bias could influence the present results. The short observation period did not allow detecting statistically significant changes of the cIMT overall or between the phytoestrogen and placebo recipients in the early postmenopausal group. Therefore, the effect of phytoestrogen therapy on atherosclerosis progression in the early postmenopausal period could not be evaluated. However, in the late postmenopausal group of women aged >60 years, the rate of cIMT progression was higher and allowed revealing statistically significant differences after 2 years of observation. In this subgroup, the alleviating effect of phytoestrogens in cIMT increase could be assessed. During the follow-up period, the hormonal status and the rate of cIMT progression was expected to change at least in some study participants. These changes may have influenced the patients’ response to the study therapy, which is, however, difficult to estimate. In addition, the study did not account for the possible variance between the study participants in the intake of phytoestrogens from dietary sources, assuming that this variance would be corrected by the randomization.

The preliminary results of Karinat^®^ effects on atherosclerosis progression have been published previously [35]. According to the inclusion criteria for the late postmenopausal subgroup (more than 5 years after menopause), women that participated in the study were more than 10 years after menopause, that allowed evaluating cIMT progression rates and effect of phytoestrogens in the late postmenopausal period. In the KEEPS study, cIMT was measured annually for 4 years postmenopause in women treated with HT or placebo and further for 3 years after discontinuation of treatment. The results of the KEEPS trial showed no difference in annual cIMT increase during the on-treatment period in comparison with post-treatment period in the overall group, and in placebo and hormone replacement therapy groups separately. Nevertheless, these results were limited by the small number of study participants that underwent cIMT monitoring in the post-treatment period [36].

The effectiveness of Karinat^®^ for treatment of manifestations of the climacteric syndrome in early postmenopausal women has been demonstrated in another study conducted by our group [22]. Similar results were also demonstrated in the KEEPS study: HT was beneficial to manage climacteric symptoms such as hot flashes, sexual function, and sleep disturbances, but had no effect on cardiovascular risk amelioration [31]. The obtained data suggest that phytoestrogen therapy may have a positive effect on the development of atherosclerosis after menopause. Phytoestrogen preparations can be used as a long-term treatment because of the virtual absence of side effects. One of the current concerns in the development of therapeutic strategies for late postmenopausal women is the association of HT with the development of breast cancer [37]. However, phytoestrogen therapy appears to be rather safe in this regard. Moreover, data on some anticancer effects of phytoestrogens are available, which makes these agents interesting for long-term therapy of postmenopausal women [38,39]. Cardioprotective effects of phytoestrogens are a very important issue of last-decades studies. Several cross-sectional studies show that subjects characterized by higher daily isoflavone consumption have significantly lower cIMT levels [40,41,42]. In prospective studies, the ameliorating effect of flavonoids on cIMT progression was also demonstrated [43,44]. However, longer monitoring of cIMT progression in postmenopausal women is required to draw final conclusions. In addition, a previous study reported that no association of cIMT progression and cardiovascular events was shown [45], but despite of the fact that clinical manifestations of CVD can be weakly correlated, the absence of association between atherosclerosis progression and CVD cannot be considered as proven.

## 5. Conclusions

The present study demonstrated no statistically significant cIMT increase in early postmenopausal women. In this study, the early postmenopausal population was not a suitable cohort for the evaluation of atherosclerosis progression and anti-atherosclerotic effects of phytoestrogen preparations. The significant cIMT increase was shown in late postmenopausal women that allowed revealing a potential beneficial effect of phytoestrogens with regard to atherosclerosis progression in common carotid artery.

## Figures and Tables

**Table 1 biology-09-00048-t001:** Clinical and laboratory data of early postmenopausal study participants at baseline.

Characteristics	Overall *n* = 315	Phytoestrogen *n* = 161	Placebo *n* = 154	Between-Groups Difference, *p*-Value
Age, years	51.8 ± 2.0	51.2 ± 2.4	52.0 ± 1.8	0.732
FSH, Ui/L	91.4 ± 10.1	89.5 ± 11.4	92.3 ± 10.3	0.343
Body mass index, kg/m^2^	26.1 ± 1.8	26.3 ± 2.1	25.6 ± 1.4	0.621
Systolic BP, mmHg	121 ± 11	117 ± 12	123 ± 8	0.512
Diastolic BP, mmHg	79 ± 4	79 ± 8	79 ± 3	0.891
Total cholesterol, mmol/L	5.3 (±SD 0.7)	5.4 (±SD 0.7)	5.3 (±SD 0.6)	0.637
Triglycerides, mmol/L	1.3 (±SD 0.7)	1.4 (±SD 0.8)	1.2 (±SD 0.4)	0.312
HDL cholesterol, mmol/L	2.5 (±SD 1.1)	2.6 (±SD 0.8)	2.3 (±SD 0.9)	0.059
LDL cholesterol, mmol/L	2.3 (±SD 0.9)	2.5 (±SD 0.9)	2.1 (±SD 0.8)	0.411

**Table 2 biology-09-00048-t002:** Clinical and laboratory data of late postmenopausal study participants at baseline.

Characteristics	Overall *n* = 231	Phytoestrogen *n* = 119	Placebo *n* = 112	Between-Groups Difference, *p*-Value
Age, years	61.4 ± 2.1	60.1 ± 2.1	62.6 ± 2.4	0.542
Time after menopause, years	10.4 ± 10.1	9.6 ± 1.1	12.0 ± 3.2	0.098
Body mass index, kg/m^2^	27.2 ± 1.4	27.3 ± 2.1	27.0 ± 0.6	0.876
Systolic BP, mmHg	128 ± 14	127 ± 9	128 ± 3	0.311
Diastolic BP, mmHg	83 ± 4	79 ± 8	84 ± 2	0.291
Total cholesterol, mmol/L	5.7 (±SD 0.7)	5.8 (±SD 0.9)	5.6 (±SD 0.6)	0.437
Triglycerides, mmol/L	1.5 (±SD 0.7)	1.5 (±SD 0.6)	1.6 (±SD 0.3)	0.753
HDL cholesterol, mmol/L	2.2 (±SD 1.0)	2.1 (±SD 0.9)	2.3 (±SD 1.0)	0.225
LDL cholesterol, mmol/L	2.9 (±SD 0.9)	2.7 (±SD 0.7)	2.9 (±SD 0.8)	0.269

**Table 3 biology-09-00048-t003:** Carotid intima media thickness (IMT) changes from baseline to 2-year follow-up visit in the early postmenopausal subgroup.

	Overall *n* = 306	Phytoestrogens *n* = 155	Placebo *n* = 151	Between-Groups Difference, *p*-Value
cIMT at baseline, mm	0.784 ± 0.098	0.765 ± 0.088	0.798 ± 0.103	0.285
cIMT at 1st year of follow-up, mm	0.790 ± 0.088	0.771 ± 0.097	0.805 ± 0.099	0.346
cIMT at 2nd year of follow-up, mm	0.798 ± 0.079	0.778 ± 0.091	0.809 ± 0.079	0.278
Annual change, mm	0.007 ± 0.027	0.007 ± 0.028	0.006 ± 0.033	0.238
Absolute 2-year change, mm	0.013 ± 0.058	0.013 ± 0.054	0.012 ± 0.065	0.471
Within-group 2-year change, *p*-value	0.342	0.433	0.401	

**Table 4 biology-09-00048-t004:** Carotid IMT changes from baseline to 2-year follow-up visit in the late postmenopausal subgroup.

	Overall *n* = 231	Phytoestrogens *n* = 119	Placebo *n* = 112	Between-Groups Difference, *p*-Value
cIMT at baseline, mm	0.844 ± 0.131	0.849 ± 0.127	0.842 ± 0.189	0.112
cIMT at 1st year of follow-up, mm	0.859 ± 0.099	0.858 ± 0.107	0.862 ± 0.101	0.622
cIMT at 2nd year of follow-up, mm	0.875 ± 0.097	0.870 ± 0.106	0.879 ± 0.097	0.538
Annual change, mm	0.014 ± 0.017	0.011 ± 0.018	0.019 ± 0.007	0.021 *
Absolute 2-year change, mm	0.028 ± 0.033	0.021 ± 0.025	0.037 ± 0.052	0.014 *
Within-group 2-year change, *p*-value	0.054	0.074	0.001 *	

Note: *, significant difference at *p* < 0.05.

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
