# Peer review of "Carotid Atherosclerosis Progression in Postmenopausal Women Receiving a Mixed Phytoestrogen Regimen: Plausible Parallels with Kronos Early Estrogen Replacement Study"

_biology, 2020, doi:10.3390/biology9030048_

Round 1
Reviewer 1 Report
To evaluate the effect of phytoestrogen therapy on progression of intima media thickness (CIMT), this author conducted a follow up study of 315 early postmenopausal women and 231 late postmenopausal women. And this author found that, the beneficial phytoestrogen on CIMT progression was demonstrated in late postmenopausal women. I thought this study deals with interesting topic and possess newly informative knowledge. Even though, there are still same points that should be clarified before publish. I commented as following.
1. An axial resolution.
Normally, CIMT values were evaluated by eye measurements by using under B-mode ultrasonography and its axial resolution using this system is ≥0.1mm [Ref01]. The main results of present study are annual change of CIMT and absolute 2-years change of CIMT among late postmenopausal women; the annual change of CIMT (0.011±0.018 for with subjects taking phytoestrogens and 0.019±0.007 for that with placebo) and absolute 2-year change of CIMT (0.021±0.025 for with subjects taking phytoestrogens and 0.037±0.052 for that with placebo). To evaluate CIMT value, this study used dedicated software. Even though, is this software really ensured an axial resolution level ≥0.001mm? If not, those measurement values could not holds enough value to discuss. In such a case, study design should be reconsidered.
[Ref01] Yanase T, et al. Evaluation of a new carotid intima-media thickness measurement by B-mode ultrasonography using an innovative measurement software, intimascope. Am J Hypertens. 2006;19:1206-12.
2. Distribution of baseline CIMT values and annual progression values of CIMT for each group should be discussed. Sample size of present study is small. Then selection bias also could influence on present results.
3. Potential biological mechanism that underlying present results also should be discussed.
Even atherosclerosis is well known factor that is associated with cardiovascular disease, previously study reported that the progression of CIMT is not associated with cardiovascular event [Ref02]. Therefore, the meaning of preventing progression of CIMT could be lack of clinical importance when it comes to focus on preventing cardiovascular disease.
Since progression of atherosclerosis is result of aggressive endothelial repair and aggressive endothelial repair might be stimulated by endothelial injury, phytoestrogen might have beneficial influence on preventing endothelial injury. If blood pressure level could influenced on present results, the phytoestrogen also could be influenced on blood pressure level.
Then additional analysis, that reveal the influence of phytoestrogen on blood pressure (Hypertension) also could be informative.
[Ref02]
Lorenz MW, et al. Carotid intima-media thickness progression to predict cardiovascular events in the general population (the PROOG-IMT collaborative project): a meta-analysis of individual participant data. Lancet 2012:379(9831):2053-62.
4. Why present major findings were observed limited to late postmenopausal women also should be discussed.
Low grade inflammation and increased levels of oxidative stress which associated with aging is well known factor that associated with vascular aging [Ref03]. Then the fact that the main results in preset study is observed only among late postmenopausal women could be explained by process of age-related endothelial injury.
[Ref03] Ungvari Z, et al. Mechanism of vascular aging. Cir Res 2018;123:849-867.
5. Drinking status and smoking status are also known classical cardiovascular risk factors. Therefore those factors also should be considered.
Author Response
Dear Reviewer 1,
On behalf of my coauthors, I express my sincere thanks for valuable suggestions and constructive input to improve the quality of this manuscript. We have revised the manuscript based upon your recommendations. A “point-by-point” response to each comment is attached.
Please see the attachment.

Reviewer 2 Report
This is a randomized double-blind, placebo-control, clinical trial investigating the effect of daily intake of phytoestrogen preparation on atherosclerosis in younger (40 to 55 years) and older (60 to 69 years) postmenopausal women. The study is approved by the IRB and the investigators are well-experience in conducting clinical trials.
My major concerns are listed below:
- There is no assessment of compliance during the trial period. This might have introduced serious bias into the study.
- RE: the inclusion criteria: For the younger women, it is imperative to know their serum estradiol level to define them postmenopausal, which was not considered in this trial. These women could be perimenopausal, which is very different than postmenopausal.
- Again, according to the inclusion criteria, women were included based on their age, not time since menopause. Therefore, it is not appropriate to refer to the groups as early-late; younger vs older would be more appropriate.
- RE: the statistical analysis approach: A mixed effects model evaluating the rate of change over 2 years for each group would be more appropriate.
- According to the results, the difference in average cIMT progression rate is different in older women but not in younger group. This differential effect should be test by an appropriate interaction term in the model.
- The discussion does not delve into the explanation as to why the effects of phytoestrogen in cIMT progression was observed only among older women.
Minor comments:
- There is no mention of the study design in the abstract. I think it is critical to mention that it was a randomized double-blind, placebo-control, clinical trial.
- There is a typo in the 2nd sentence of the abstract: last should be late.
- It is not clear if there were any surgically menopaused women included in the study.
Author Response
Dear Reviewer 2,
On behalf of my coauthors, please accept my sincere thanks and gratitude for careful perusal and critical review of our manuscript. Adequate care has been taken to accommodate each and every suggestion. An itemized, “point-by-point” reply to all the comments is attached separately where we have clearly presented our specific response and additions, deletions and/or modifications that have been made in the revised text.
Please see the attachment.
Sincerely Yours,
Tatiana Kirichenko, corresponding author

Round 2
Reviewer 1 Report
I appreciate this author’s great effort to improve this manuscript. I’m almost satisfied with this revised version of manuscript. Only one matter that what I thought important. That is about an axial resolution. This author commented as following.
The M'Ath software allows to make measurements with a precision < to 0.01, as the value of IMT is based on the mean of 150 measurements on a 10mm length of the far wall with a final value as 0.00X.
However, I could not found this sentence in present manuscript. Please add this sentence with same reference in text. Furthermore, in this revised manuscript this author used three decimal places. The sentence “The M'Ath software allows to make measurements with a precision < to 0.01” allows to use 0.01 mm level but not 0.001 level.
Author Response
Dear Reviewer 1,
Regarding your last question:
However, I could not found this sentence in present manuscript. Please add this sentence with same reference in text. Furthermore, in this revised manuscript this author used three decimal places. The sentence “The M'Ath software allows to make measurements with a precision < to 0.01” allows to use 0.01 mm level but not 0.001 level.
We can comment the following:
We have used the recommendations on reporting statistics by Lang and Secic (1), reference was added in article, see below. Depending on the accuracy of the tools we employ in research, each variable is measured within a certain degree of precision. However, there is the unresolved issue on reporting the statistics of the mean values in scientific articles. There is no consensus on this issue (2). For example, some references suggest that in reporting statistics (eg, means and standard deviations) not to use precisions higher than the accuracy of the measured data (3). At the same time, most of researchers recommend to use one decimal place more than the precision used to measure the variable (1,4); and, some mention that although means should not be reported to no more than one decimal place more than that of the raw data, SDs may need to be reported with an extra decimal place (5).
The manual on M’Ath software (https://www.iimt.fr/wp-content/uploads/2018/07/Math_3_2_1ENmanual05052018.pdf) reports the results of cIMT measurement with a precision of 0.001 mm, see example of measurement at page 34. Therefore, upon calculation of mean values for the whole sample, we considered correct to use three decimal places, which is not controversial to all statisticians’ recommendations. Moreover, our paper reports statistics just in line with other researchers, who used this dedicated software (6).
- Lang TA, Secic M. How to report statistics in medicine: annotated guidelines for authors, editors, and reviewers. 2nd ed. Philadelphia: American College of Physicians; 2006.
- Habibzadeh F, Habibzadeh P. How much precision in reporting statistics is enough?. Croat Med J. 2015;56(5):490–492.
- Priebe HJ. The results. In: Hall GM, editor. How to write a paper. 3rd ed. London: BMJ Publishing Group; 2003. p. 22-35.
- Peat J, Elliot E, Baur L, Keena V. Scientific writing: easy when you know how. London: BMJ Publishing Group; 2002.
- Altman DG, Gore SM, Gardner MJ, Pocock SJ. Statistical guidelines for contributors to medical journals. BMJ. 1983;286:1489–93.
- Touboul PJ, Vicaut E, Labreuche J, Belliard JP, Cohen S, Kownator S, Pithois-Merli I; Paroi Artérielle et Risque Cardiovasculaire Study Investigators. Design, baseline characteristics and carotid intima-media thickness reproducibility in the PARC study. Cerebrovasc Dis. 2005;19(1):57-63.
Thus, according to your recommendations the following changes were made in article:
Lines 137-139: “using dedicated software package M’Ath 3.1 (Metris, SRL France), which allows to make single measurements with a precision of <0.07 mm; since the mean value of cIMT is calculated from 100 to 150 measurements on a 10 mm length of the far wall, the resulting precision of individual result accounts for <0.01 mm [26].” was added in the Methods section with the appropriate reference to the software manual. “We have used the recommendations on reporting statistics by Lang and Secic [28].” was added at lines 153-154 with the reference #28.
Sincerely Yours,
Tatiana Kirichenko, corresponding author.
Reviewer 2 Report
The authors responded to all my major comments. They have included suggested explanations, and clarifications as requested.
Author Response
Dear Reviewer 2,
On behalf of my coauthors, I once again express my sincere thanks for your time and efforts in improving of our manuscript and your favorable decision.
Sincerely Yours,
Tatiana Kirichenko, corresponding author.